# STIMULATE-ICP: A pragmatic, multi-centre, cluster randomised trial of an integrated care pathway with a nested, Phase III, open label, adaptive platform randomised drug trial in individuals with Long COVID: A structured protocol

**Denise Forshaw**[1], **Emma C. Wall**[2,3,4], **Gordon Prescott**[1], **Hakim-Moulay Dehbi**[5], **Angela Green**[6], **Emily Attree**[7], **Lyth Hismeh**[7], **William D. Strain**[8,9], **Michael G. Crooks**[10,11], **Caroline Watkins**[1], **Chris Robson**[12], **Rajarshi Banerjee**[13], **Paula Lorgelly**[14,15], **Melissa Heightman**[16,17], **Amitava Banerjee**[18,19,20]*, **the STIMULATE-ICP trial team**[¶]

1 Lancashire Clinical Trials Unit, University of Central Lancashire, Preston, United Kingdom, 2 Francis Crick Institute, London, United Kingdom, 3 National Institute for Health Research (NIHR) University College London Hospitals (UCLH) Biomedical Research Centre, London, United Kingdom, 4 Division of Infection and Immunity, University College London, London, United Kingdom, 5 Comprehensive Clinical Trials Unit, University College London, London, United Kingdom, 6 Department of Physiotherapy, Hull University Teaching Hospitals NHS Trust, Hull, United Kingdom, 7 STIMULATE-ICP PPI Team, 8 University of Exeter Medical School, Exeter, United Kingdom, 9 Academic Department of Healthcare for Older People, Royal Devon and Exeter NHS Foundation Trust, Exeter, United Kingdom, 10 Academic Respiratory Medicine, Hull University Teaching Hospitals NHS Trust, Hull, United Kingdom, 11 Hull York Medical School, Institute of Clinical and Applied Health Research, University of Hull, Hull, United Kingdom, 12 Living With, London, United Kingdom, 13 Perspectum, Oxford, United Kingdom, 14 School of Population Health, University of Auckland, Auckland, New Zealand, 15 Department of Applied Health Research, University College London, London, United Kingdom, 16 Department of Respiratory Medicine, University College London Hospitals NHS Trust, London, United Kingdom, 17 NHS England, London, United Kingdom, 18 Institute of Health Informatics, University College London, London, United Kingdom, 19 Department of Cardiology, Barts Health NHS Trust, London, United Kingdom, 20 Department of Cardiology, University College London Hospitals NHS Trust, London, United Kingdom

¶ Membership of the STIMULATE-ICP trial team is provided in the Acknowledgments.
* ami.banerjee@ucl.ac.uk

**Data Availability Statement:** The protocol does not report results and no further data are available.

## Abstract

### Introduction

Long COVID (LC), the persistent symptoms ≥12 weeks following acute COVID-19, presents major threats to individual and public health across countries, affecting over 1.5 million people in the UK alone. Evidence-based interventions are urgently required and an integrated care pathway approach in pragmatic trials, which include investigations, treatments and rehabilitation for LC, could provide scalable and generalisable solutions at pace.

### Methods and analysis

This is a pragmatic, multi-centre, cluster-randomised clinical trial of two components of an integrated care pathway (Coverscan™, a multi-organ MRI, and Living with COVID

Deidentified research data will be made publicly available when the study is completed and published.

**Funding:** This trial is part of a National Institute for Health Research (NIHR: COV-LT2-0043)-funded programme (Figure 2)(17), with epidemiologic and mixed methods studies(18), including care inequalities and transferability to other LTCs (IRAS 303958) and this complex intervention trial (IRAS 1004698). Funding for the transport, storage, processing and biobanking of blood and MRI is supported by Perspectum Ltd. The funders had and will not have a role in study design, data collection and analysis, decision to publish, or preparation of the manuscript. The views and opinions expressed therein are those of the authors and do not necessarily reflect those of the NIHR, NHS, the Department of Health and Social Care, or the sponsor.

**Competing interests:** We have read the journal's policy and the authors of this manuscript have the following competing interests: WDS holds research grants from Bayer, Novo Nordisk and Novartis and has received speaker honoraria from AstraZeneca, Bayer, Bristol-Myers Squibb, Merck, Napp, Novartis, Novo Nordisk and Takeda, outside the submitted work. WDS is supported by the NIHR Exeter Clinical Research Facility and the NIHR Collaboration for Leadership in Applied Health Research and Care (CLAHRC) for the South-West Peninsula. MGC has received honoraria, fees for advisory boards, and non-financial support from AstraZeneca, BI, Chiesi, GSK, Novartis, and Pfizer and grants from AstraZeneca, BI, Chiesi, and Pfizer. CR is director of Living With, which developed Living With COVID Recovery™ and other digital health interventions used in healthcare systems including the NHS. RB is CEO of Perspectum, which developed Coverscan™. AB has received research funding from NIHR, British Medical Association, Astra Zeneca, National Institute of Aging and European Union. AB is a Trustee of Long COVID SoS. All other authors report no competing interests. This does not alter our adherence to PLOS ONE policies on sharing data and materials.

Recovery™, a digitally enabled rehabilitation platform) with a nested, Phase III, open label, platform randomised drug trial in individuals with LC. Cluster randomisation is at level of primary care networks so that integrated care pathway interventions are delivered as "standard of care" in that area. The drug trial randomisation is at individual level and initial arms are rivaroxaban, colchicine, famotidine/loratadine, compared with no drugs, with potential to add in further drug arms. The trial is being carried out in 6–10 LC clinics in the UK and is evaluating the effectiveness of a pathway of care for adults with LC in reducing fatigue and other physical, psychological and functional outcomes at 3 months. The trial also includes an economic evaluation which will be described separately.

## Ethics and dissemination

The protocol was reviewed by South Central—Berkshire Research Ethics Committee (reference: 21/SC/0416). All participating sites obtained local approvals prior to recruitment. Coverscan™ has UK certification (UKCA 752965). All participants will provide written consent to take part in the trial. The first participant was recruited in July 2022 and interim/final results will be disseminated in 2023, in a plan co-developed with public and patient representatives. The results will be presented at national and international conferences, published in peer reviewed medical journals, and shared via media (mainstream and social) and patient support organisations.

## Trial registration number

ISRCTN10665760.

## Introduction

The syndrome of Long COVID (LC), defined by persistent post-COVID-19 symptoms ≥12 weeks [1], has affected over 2 million people in the UK alone [2], necessitating health system responses despite incomplete disease characterisation. While many symptoms are reported by individuals with LC, fatigue is the most frequent and most commonly reported to limit quality of life [3, 4]. Among other symptoms are chest pain, 'brain fog', dyspnoea, headache, dizziness, palpitations, and sleep disturbances [5]. Unlike acute COVID-19 [6], predictors of poor outcomes and effective management are not yet established for LC [7–9]. Integrated care pathways are structured, multidisciplinary plans in care of specific conditions [10], offering coordination across specialties, investigations, treatment, and rehabilitation, as well as opportunities for real-time, iterative improvements in service design and delivery [11–13].

There is variation in access to referral and care, and poor patient experience for LC [14]. With stretched resources during the pandemic, rationalisation of investigation and rehabilitation across diseases is crucial. In LC, non-hospitalised individuals were more symptomatic, more likely to have psychological impact, and less likely to be fit for work than post-hospitalised individuals. Up to 70% of individuals have mild organ impairment using Coverscan™, a multi-organ MRI protocol [15]. In the context of variable waiting times, resources and practice (with frequent repeating of investigations and referrals to multiple specialists [3]), an early supported, multi-organ investigation strategy like this may direct management, as well as better defining the syndrome, with relevance to other long-term conditions. A negative finding could rule out organ involvement and other disease early, reassuring clinicians and patients, rather

than organising multiple investigations with multiple specialists via multiple referrals. By flagging organ dysfunction, early, appropriate specialist referral and management may be initiated. There is no evidence-based treatment or rehabilitation, with few clinical trials to-date, particularly in non-hospitalised individuals, or considering overall care pathways. Although digital rehabilitation platforms (e.g. Living with COVID Recovery™) may be scalable, particularly with strained rehabilitation services, and already being used in some centres, they are not yet evaluated in trials [16]. Despite roll-out of 90 centrally funded LC clinics in England, ICPs are neither coordinated nor evaluated. Trials of integrated care pathways could inform diagnosis, care, public health, policy planning, resource allocation and budgeting.

Antihistamines (e.g., loratadine and famotidine), colchicine and rivaroxaban are examples of medications being widely used among patients and health professionals with rationale for use (**S1 Appendix**), but without proven efficacy. Platform trials have been used at scale to evaluate therapies in acute COVID-19 [17], but not in LC. Since underlying causes are unknown and subtypes poorly defined, trials should test drugs across all individuals with LC, irrespective of individual symptoms or investigation findings, to allow multiple secondary sub-group analyses. Optimal outcomes for LC are yet to be fully defined. Therefore, trials should evaluate a wide range of potential physical, psychological and functional outcomes. We report the protocol for the STIMULATE-ICP (Symptoms, Trajectory, Inequalities and Management: Understanding Long COVID to Address and Transform Existing Integrated Care Pathways) trial (**Fig 1**).

## Objectives

For individuals with LC, to evaluate the:

1. effect of "integrated care" versus "usual care".

2. clinical efficacy of

    a. individual components of an integrated care pathway

    b. potential drug therapies within a nested drug trial.

3. pathophysiology, trajectory and outcomes, including healthcare utilisation (**Tables 1 and 2**)

## Methods and analysis

### Design

STIMULATE-ICP is a pragmatic, cluster-randomised trial of two components of an integrated care pathway with a nested, Phase III, open label platform randomised drug trial in individuals with LC. It is responsive to local resources and needs, *reactive* to changing pandemic and policy contexts and *iterative*, depending on ongoing results, as part of a National Institute for Health Research (NIHR: COV-LT2-0043 [18]) -funded programme (**Fig 2**), with epidemiologic and mixed methods studies, including care inequalities and transferability to other long-term conditions [19]; IRAS 303958 [20]) and this complex intervention trial (IRAS 1004698).

### Setting

This multi-centre trial involves Primary Care Networks and LC Clinics in 6–10 different English areas, covering diversity of geography, demography and clinical service design. Management is by Lancashire Clinical Trials Unit and sponsorship by University College London.

| | Procedures | Referral to LCC and /or Coverscan™* | First Long COVID Clinic Visit | Baseline/ Visit | Treatment / Assessments | Follow Up / Assessments |
|---|---|---|---|---|---|---|
| | | | | STIMULATE-ICP Trial Schedule of Assessments | | |
| Visiting window | | | Day 0 | No later than 7-10 days post clinic* | 12 weeks (+7days)*** | 24 weeks (+7days)**** |
| Visits No | | | 0 | 1 | 2 | 3 |
| | Informed consent | | X* | | | |
| Patient's History | Demographics | | | X | | |
| | Medical history | | | X | | |
| | Concomitant Medications | | | X | | |
| Physical examination | Height, Weight, Temperature | | | X | | |
| | Resting Pulse and BP | | | X | | |
| Laboratory Tests[1] | Full blood count | | | X | | |
| | Liver function test | | | X | | |
| | Estimated glomerular filtration rate | | | X | | |
| | Coagulation screening | | | X | | |
| | Urine pregnancy test[5] | | | X | | |
| | COVID-19 Antibody | | | X | | |
| Sample Collections | Samples of blood for biobank storage at Perspectum, Genomics, proteomics, metabolomic & Lipidomic, functional T cell and live virus Neutralisation antibody, Endocrine analysis | | | X | X[6] | |
| Cardiovascular | (electrocardiogram, echocardiogram, Holter Monitor, Stress Electrocardiogram)[1] | | | X | | |
| Pulmonary | chest radiograph or cardiac magnetic resonance imaging, CT pulmonary angiogram[1] | | | X | | |
| | 6 Minute Walk Test, 1-minute Sit to Stand test, | | | X | x | x |
| | Pulmonary function test (FeNo)[1] | | | x | | |
| CNS | Magnetic resonance imaging of brain[1] | | | X | | |
| | Tilt Table Test | | | X | | |
| Whole Body | Coverscan™* ** | X | | | | |
| Inclusion and Exclusion | Eligibility assessment | | | X | | |
| | Randomisation | | | X | | |
| IMP (Investigational Medical Product) | Dispensing of trial IMP[3] | | | X | | |
| | Accountability | | | | X | |
| Patient Reported Outcome Assessments | FAS Assessment | | | X | X | X |
| | MRC Dyspnoea Score | | | X | X | X |
| | General Anxiety Disorder Questionnaire- 7 (GAD-7) | | | X | X | X |
| | The Primary Care Evaluation of Mental Disorders Patient Health Questionnaire (PHQ-9) | | | X | X | X |
| | EQ-5D-5L (EUROQOL-5 domain- 5 level) | | | X | X | X |
| | Perceived Deficit Questionnaire) PDQ-5 | | | X | X | X |
| | SF12 | | | X | X | X |
| | Cognitive Failure Questionnaire (CFQ) [4] | | | X | X | X |
| | Modified Work and Social Adjustment Scale (WSAS) | | | X | X | X |
| Safety Assessments | Adverse Event Reporting | | | | X | X[b] |
| Concomitant medication | Concomitant medication review | | | | X | X[7] |

**Fig 1. Schedule of assessments in the STIMULATE-ICP trial.** Consent for data collection only may take place before first clinic visit and up to 10 days post clinic. Consent for drug arm must be at clinic visit (to assess eligibility based on clinic record) or in the 7 days post clinic. This is to align the administration on IMP with follow-up data collection and clinic pathway. Consent for research blood to be taken at either time point as per patient choice. **Coverscan as usual care for allocated clusters. ***: Assessment 1 is 12 weeks from baseline visit date (date of randomisation or first dose of study IMP) or consent date at baseline visit (if only consented to data collection). ****: Follow up assessment 24 weeks from baseline visit date (date of randomisation or first dose of study IMP) or consent date at baseline visit (if only consented to data collection). 1: If requested by the treating physician at Long COVID Clinics as part of their routine care. 2: Only if recruited from a site randomised to Coverscan™ cluster. 3: If randomised to receive the trial drug. 4: if the participant scores ≥3 on PDQ5 questionnaire. 5: For female participants of childbearing potential randomised to drug arm of the trial.

6: blood samples collected for functional T cell and live virus neuralisation assays from London Participants only at baseline and 12 weeks. All other participating centres will collect blood for external laboratory analysis once at baseline visit. Permission will be sort from all participants to recontact to participate in further testing due to any changes to the study on the basis of emerging evidence/ results. 7. concomitant medications will be reviewed at 12 and 24 weeks visit is only for participants o the nested drug trial. b. Adverse events will be collected at the 24 week visit for participants on the nested drug trial only.

## Intervention

In this trial, "integrated care" comprises two integrated care pathway interventions, cluster-randomised at Primary Care Network level: Coverscan™ (community based, multi-organ MRI with clinical decision support)(15) and Living with COVID Recovery™ (enhanced community based, digitally enabled rehabilitation)(16). The Coverscan will have a standardised report. For the first 100 scans across sites, there will be report by an independent radiologist, and a virtual, weekly multidisciplinary meeting to discuss any cases and potential implications of findings, management and/or referral to specialists. We did not have or develop an algorithmic management approach (i.e. clinical decision support) or a "manual" because the organ dysfunction in Long Covid is still being defined and a manual would involve approaches to any possible finding in lungs, heart, liver, pancreas and kidneys, which is not feasible, at present.

"Usual care" comprises usual investigations (e.g. routine blood tests, electrocardiogram, chest radiograph or exercise tolerance test); and self-managed rehabilitation with online resources (Your COVID Recovery™: https:\\www.yourcovidrecovery.nhs.uk) in the LC clinic pathway, which is standard of care.

In a nested, open label, adaptive platform trial, a 12-week course of one of three drug arms will be evaluated (individual level randomisation): famotidine+loratadine, colchicine and rivaroxaban, compared with no drugs.

## Sites

This is a pragmatic trial, comparing two integrated care pathway interventions to usual care, ensuring that research does not impact/change workload for already over-stretched healthcare resources, whilst ensuring equity of access to care.

## Randomisation

Given current COVID-related health system pressures, individual level randomisation for integrated care pathway interventions (Coverscan™ and Living with COVID Recovery™) at

**Table 1. Research question in STIMULATE-ICP.**

| |
|---|
| **Over-arching research question:** |
| What is the effectiveness of an integrated care pathway for individuals with Long Covid, compared with usual care? |
| **Population:** |
| Individuals ≥18 years with ≥4 weeks post-COVID symptoms (in line with UK clinical guidance and policy), based on clinician diagnosis, newly referred to participating Long COVID clinics. |
| **Intervention:** |
| *Investigation*: Multi-organ MRI prior to Long Covid clinic assessment (Coverscan™) |
| *Treatment*: Multiple drug arms, starting with famotidine/loratadine, colchicine and rivaroxaban |
| *Rehabilitation*: Digitally enabled rehabilitation (Living with COVID Recovery™) |
| **Comparator:** |
| *Investigation*: Usual care |
| *Treatment*: No drug |
| *Rehabilitation*: Usual care |
| **Outcomes:** |
| *Primary*- Fatigue on the fatigue assessment scale at 12 weeks. |
| *Secondary*- Multiple assessments of health-related quality of life, mental health (including depression), work and social adjustment, physical function, organ impairment and healthcare utilisation at 12 and 24 weeks. |

**Table 2. Summary of the primary and secondary objectives and outcome measures.**

| Objectives | Outcome Measures | Timepoint(s) of evaluation of this outcome measure |
|---|---|---|
| **Primary Objective**<br>Evaluation of the effect of "integrated care" with combinations of multi-organ MRI (Coverscan™) and clinical decision support, and digitally enabled community rehabilitation (Living with COVID Recovery™) versus "usual care" [Usual Investigations; and Your COVID Recovery ™ self-management website] on Fatigue Assessment Scale at 12 Weeks from the date of enrolment in the trial. | FAS | Baseline, 12 and 24 weeks |
| **Secondary Objectives**<br>The overall clinical efficacy of the individual components of the integrated care pathway including potential therapies.<br>1. Effect of the ICP on mean measured health related quality of life, mental health (including depression), work and social adjustment, physical function, organ impairment and healthcare utilisation, through patient reported outcomes.<br>2. To determine if pre-specified sub-groups of patients, either grouped by clinical symptom cluster or imaging findings benefit from either early investigation with COVERSCAN™, IMPs or rehabilitation. | 1. FAS at 24 weeks<br>2. Health related quality of life (EQ-5D-5L)<br>3. Mental health (GAD-7)<br>4. Depression (PHQ-9)<br>5. MRC Dyspnoea Score<br>6. Perceived Deficit Questionnaire (PDQ-5)<br>7. Work and Social Adjustment Scale (WSAS) [Question 4 from Productivity Cost Questionnaire (iPCQ) for absenteeism and Question 8 from iPCQ for presenteeism added]<br>8. Short Form Questionnaire (SF-12)<br>9. Cognitive Failure Questionnaire (CFQ) if a patient scores 3 or more on PDQ5 (patients receive an email to complete this questionnaire online via a secure password and patient ID number)<br>10. Organ impairment and healthcare utilisation<br>11. Cost-effectiveness of ICP<br>12. Process outcomes for different ICP components | Baseline, 12 and 24 weeks |
| 3. To characterise the pathophysiology and trajectory of long COVID | Blood investigations (e.g., genomic, proteomic, metabolomic/ lipidomic, functional T cell and live virus neutralisation assays and endocrine analysis) | Baseline |
| | Linked electronic health record data to monitor healthcare utilisation and outcomes. | 12 months |

General Practitioner level is not feasible and could potentially increase waiting times in other areas of care, which could affect recruitment. Unlike individual randomisation, cluster randomising at primary care network level allows 50–50 chance of allocation to both Coverscan™ and Living with COVID Recovery™, maintaining access across the total number of referred individuals and equipoise at clinic level. Randomisation will use demographic and socioeconomic data available for each primary care network, ensuring equity of access through deprivation indexes of GP postcodes. Clustering at primary care network level allows Coverscan™ and/or Living with COVID Recovery™ to be provided as uplift to usual care in allocated primary care networks, becoming integrated care within that primary care network without the need for individual consent. Access to and collection of Coverscan™ and Living with COVID Recovery™ data for research will be by individual consent on study entry (by the research staff). Cluster randomisation (2x2 factorial design) provides four groups:

a. Coverscan™ + Living with COVID Recovery™ App.

b. Coverscan™ + Usual Care Self-Management Rehabilitation.

c. No Coverscan™ + Living with COVID Recovery™ App.

d. No Coverscan™ + Usual Care Self- Management Rehabilitation.

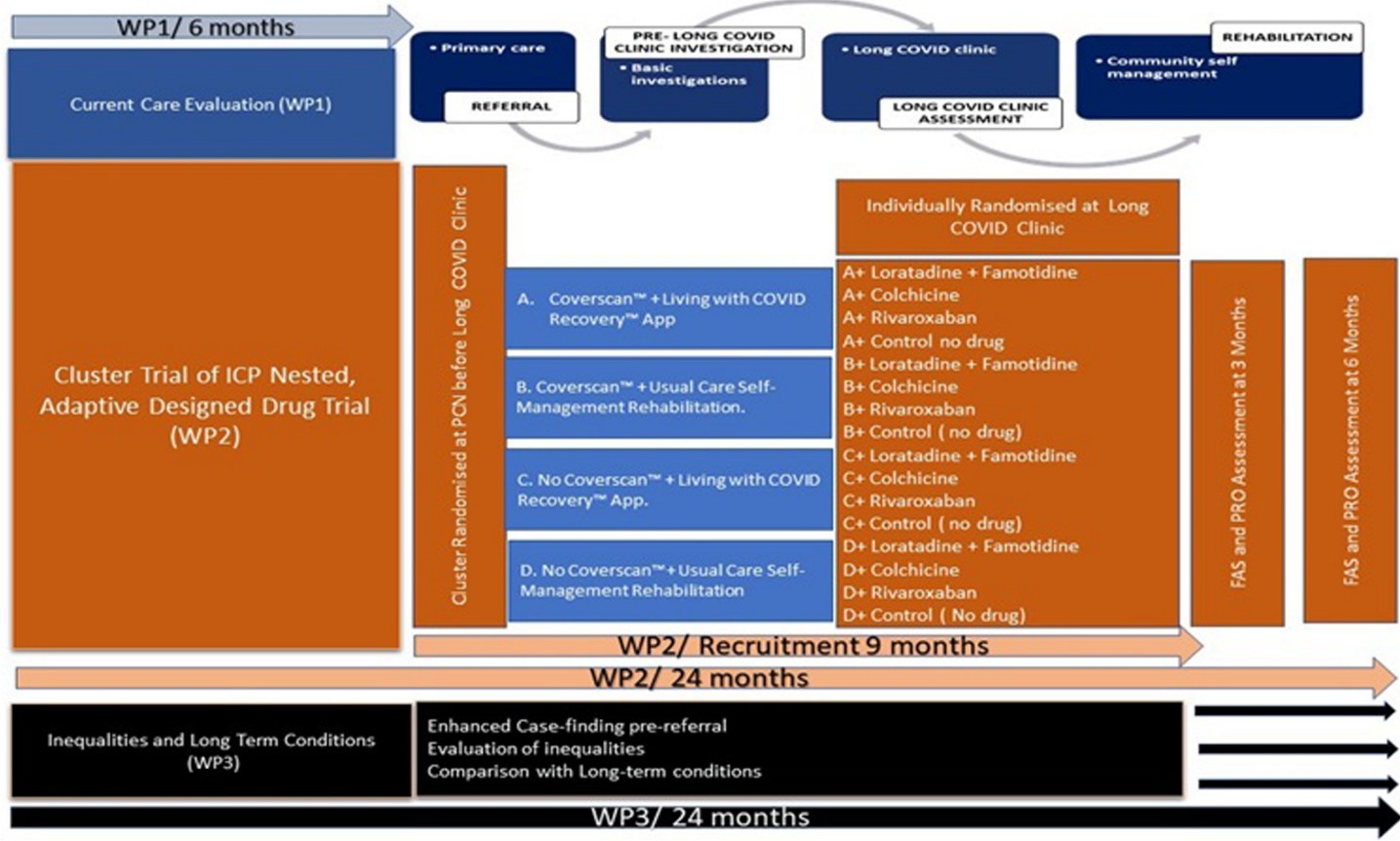

**Fig 2. The STIMULATE-ICP programme showing the interconnections between the Trial and other work packages.** Abbreviations: COVID: COrona Virus Disease; FAS: Fatigue Assessment Scale; PCN: Primary Care Network; PRO: Patient-Reported Outcome; WP: Work Package.

At the LC clinic, individuals eligible for the drug study will be randomised to one of 4 arms:

1. Famotidine + Loratadine

2. Colchicine

3. Rivaroxaban

4. No Drug

Other drugs may be added as additional or replacement arms for new patients at predetermined time points during the trial (6 months and 12 months), dependent on interim analysis of trial data with consideration given to updated recommendations from an independent advisory board.

Individuals who are ineligible for the drug study will invited to participate for data collection, which will enable data collection to represent the overall LC population.

## Randomisation methods

Block randomisation for the primary care networks will be used with blocks of size 4. There are two factors in the randomisation, and within each clinic: primary care network size (categorised into 2 levels) and primary care network deprivation (categorised into 2 levels). Within each clinic, the strata of primary care networks will be arranged as larger and more deprived;

larger and less deprived; smaller and less deprived; and smaller and more deprived, before randomised allocation into the 4 cluster allocations. The randomisation ratio is 1:1:1:1 between the 4 possible treatment pathways.

Randomisation to the drug trial at the clinic will be at individual participant level using an online system (Sealed Envelope™). Block randomisation will be used with varying block sizes. There will be two factors in the randomisation within each clinic: gender (male vs female) and treatment pathway. The randomisation ratio will be equal between all drug treatment arms (including no drug: "usual care") open to recruitment at that point in the cluster trial.

## Timeline

We aim to recruit individuals over a 12-month period. The 12-week treatment duration in the drug trial and the 12-week primary outcome, fatigue, were selected to match the 12-week duration of the commissioned National Health Service (NHS) LC clinic pathway. Outcomes will also be assessed at 24 weeks. The overall trial will run over an 18-month period.

## Participants

Participants are ≥18 years with ≥4 weeks post-COVID symptoms (in line with UK clinical guidance and policy), based on clinician diagnosis, newly referred to participating LC clinics. The vast majority of people included will have had symptoms for >12 weeks, based on delays in the current care pathways and our preliminary University College London Hospital data.

## Inclusion criteria for ALL participants

- Capable of giving informed consent.

- Age 18 years and above

- Clinical Parameters: persistent signs and symptoms for a period of 4 weeks or longer in duration post-COVID-19 infection (either by test result or symptomology). Presenting at their first referral first visit to a participating LC clinic pathway.

- Able to read or understand English or have a relative/family member able to read/understand English to facilitate participation (essential for patient reported outcome measures at follow-up time points and virtual contact).

- Not enrolled in any other interventional study where study intervention/activities may affect outcome measures (individuals enrolled in purely observational studies can be included)

**S2 Appendix** lists additional Inclusion Criteria for the nested, platform randomised drug trial to be met in addition to *all above criteria*, and also details of assessment of eligibility, recruitment and withdrawal of consent.

## Assessment and follow-up

All Participants, once consented, regardless of study arm, will be invited to complete assessments at baseline, 12 and 24 weeks.

**Baseline visit.** Demographics, full medical history (from January 2020, including comorbidities and all COVID-related illnesses, treatments, and vaccinations); history of concomitant medication including current, pre- and post- COVID-19 and over-the-counter medications; and physical examination (including weight, height, oral temperature, resting pulse, and blood pressure) will be recorded.

Routine clinical blood tests performed in primary care, and following clinic assessment at point of referral, will be collected as part of trial data. Blood tests will differ by clinic, based on differences in usual care and service capacity and capability. Individuals who consent for research blood sampling and for further research will have approximately 60mls blood taken for translational sub-studies and biobanking (**S3 Appendix**). All participants who have consented for bloods will be asked to confirm if they would be willing to be contacted at 12 or 24 weeks for further blood sampling.

Standard investigations differ from clinic to clinic. To evaluate usual care, the study will not require a standardised testing strategy; but will collect any data related to individual consented participants' investigations, including Coverscan™. Examples of functional tests in some clinics are detailed in **S4 Appendix**. If the participant consents to the drug trial, a prescription or the drug will be issued and the first dose taken at the clinic (depending on the location of the recruiting site at the participating centres) or at the time of randomisation or at home for those sites using an drug delivery service (date of first dose to be confirmed via telephone by site research team). The Patient Reported Outcome Questionnaires and functional tests to be completed are listed in **S5 Appendix** and **Fig 1**.

**12-Week assessment visit.**    Patient Reported outcome measures and activities will be collected at 12 weeks (± 7 days) to record primary and secondary outcome measures. This may not be possible in all individuals due to level of fatigue. Participants will be supported by flexibility of data collection processes. Patient reported outcome measures will be collected either at the LC clinics, by post, entered by participants themselves via secure electronic link or over the phone as per participant choice. Non-patient reported data will be extracted from clinical records by site-specific research staff and entered in the trial database (listed in **S6 Appendix**).

**24-Week assessment visit.**    All participants will be followed up for 24 Weeks (± 7 days) from enrolment date by Lancashire Clinical Trials Unit staff. Participants will be given choice of completing questionnaires online using an individualised secure link, by postal return or via telephone for participants requiring support. In the event of receiving an incomplete outcome assessment, participants will be contacted by phone within 7 days of the team receiving the form (e-form or paper) to complete missing information with the individual. Data will be entered into the trial database by Lancashire Clinical Trials Unit staff. Patient Reported Outcome measures and functional assessments recorded are listed in **S7 Appendix**.

## Outcomes

Fatigue is the dominant symptom in 60% of patients and LC clinics are being commissioned to provide rehabilitation for 12 weeks or less. Fatigue Assessment Scale (FAS) is a 10-item, validated questionnaire in individuals with chronic diseases, including LC [4, 16], making it a pragmatic primary end point at 12 weeks. Five questions reflect physical fatigue and five questions (questions 3 and 6–9) reflect mental fatigue. The total score ranges from 10 to 50.

Secondary outcomes include:

1. FAS at 24 weeks

2. Health related quality of life (EQ-5D-5L) [21]

3. Mental health (GAD-7) [22]

4. Depression (PHQ-9) [23]

5. Medical Research Council Dyspnoea Score [24]

6. Perceived Deficit Questionnaire (PDQ-5) [25]

7. Work and Social Adjustment Scale (WSAS) [Question 4 from Productivity Cost Questionnaire (iPCQ) for absenteeism and Question 8 from iPCQ for presenteeism added] [26, 27]

8. Short Form Questionnaire (SF-12) [28]

9. Cognitive Failure Questionnaire (CFQ) if an individual scores 3 or more on PDQ5 (individuals receive an email to complete this questionnaire online via a secure password and patient ID number) [29, 30]

10. Organ impairment and healthcare utilisation

11. Cost-effectiveness of integrated care pathway (a separate health economic evaluation is planned [18]) and the protocol will be published separately)

12. Process outcomes for different ICP components

13. Blood investigations (e.g., genomic, proteomic, metabolomic/ lipidomic, functional T cell and live virus neutralisation assays and endocrine analyses)

14. Linked electronic health record data to monitor healthcare utilisation and outcomes.

A core outcome data set for clinical and research use in individuals with LC is being developed. Based on a NIHR-funded research programme (Post-COVID Core Outcome Set ([31])) which conducted an extensive, global Delphi process in over 1500 patients and health professionals, we included consensus physiological/clinical, life impact and recovery domains. As our trial progresses, we will collaborate to iteratively and pragmatically develop LC core outcomes.

## Data collection

All electronic and paper clinical research forms (CRFs) must be completed and signed by designated, authorised research team staff. Information from paper CRFs will be entered into an electronic CRF within the electronic trial database by the site team directly for the baseline and 12 -week assessment (where participants attend in person). For virtual clinics and where the patients are not returning to the clinic in person, participants will be given the option of completion via postal, e-link (directly into database for self-completion) or over the telephone for participants requiring support. All the CRFs will be entered into the trial database within 15 days of data received by the site staff or CTU staff as applicable. **Table 2** summarises objectives and outcome measures and when they will be collected. All data are managed in accordance with Lancashire Clinical Trials Unit standard operating procedures (DM-03 and DM-05) and a trial-specific Data Management Plan. Where data are transferred electronically this will be in accordance with the UK Data Protection Act 2018 as well as sponsor and trials unit information security and governance policies. Further details regarding data collection and the Data Management Plan are in **S8 Appendix**.

## Analysis

The trial will be analysed and reported using "Consolidated Standard of Reporting Trials" ("CONSORT") [32] and International Conference on Harmonisation E9 guidelines [33]. Baseline characteristics of all consenting participants will be reported by frequency and percentage for categorical variables, and by mean and standard deviation for continuous variables (or median and inter-quartile range for non-normally distributed data).

## Primary outcomes

Primary analysis of the trial will be a complete case analysis using all available outcomes, according to cluster randomised allocation of participant's primary care network (regardless of treatment pathways received) or the stratified randomisation of the individual participant (regardless of treatment received). The primary outcome, FAS at 12 weeks, is expected to be approximately normally distributed [34]. A multi-level model analysis will be used to evaluate effects of Coverscan™, Living with COVID Recovery™, and their interaction, on FAS at 12 weeks adjusting for baseline FAS, with clinic and primary care network as random effects.

## Effect of "integrated care" versus "usual care"

The principal analysis population for estimates of the interaction of effects of Coverscan™, Living with COVID Recovery™ will be those participants (up to 1130) who consented to data collection for the cluster RCT and were allocated to usual care (either by randomisation to no drug in the drug trial or being willing for data collection, but not eligible for the drug trial, or unwilling to be randomised into the drug trial; (Sample 1 in **Fig 2**)). This analysis will provide the most straightforward answer to the primary objective in the broadest group of participants. A similar and more pragmatic multi-level model analysis of effects of Coverscan™, Living with COVID Recovery™, and their interaction, will be applied to all participants who consent to data collection in the cluster trial regardless of participation in the drug trial. This analysis will provide an answer to the primary objective in the presence of other treatments including drug treatments and will allow comparison to principal analysis results.

## Clinical efficacy of individual components of an integrated care pathway

This analysis is similar to the prior analysis but focusing on individual effects of Coverscan™ and Living with COVID Recovery™.

## Clinical efficacy of potential drug therapies within a nested drug trial

The principal analysis population for drug vs. usual care (no drug) comparisons within the drug trial will be the participants randomised to the control treatment pathway, i.e. no Coverscan + usual care rehabilitation (Sample 2 in **Fig 3**). Participants randomly allocated to each drug arm will be compared to those randomly allocated to usual care while both that drug and usual care are options for randomisation (concurrent usual care controls). A multi-level model for FAS, adjusting for baseline, with gender as a fixed effect and clinic and primary care network as random effects, will be applied to the drug trial data. By restricting comparisons of effects of drugs vs. usual care on fatigue to those participants allocated to the treatment pathway incorporating the lowest level of intervention, and potentially the least variation in care, this will give the cleanest and least biased comparison. The total population of participants who are randomised into the drug trial will be used to study effects of drugs in combination/ interaction with the pathways.

A multi-level model for FAS, adjusting for baseline, with gender as a fixed effect and clinic and primary care network as random effects, will be applied to data from all participants of both the cluster trial and the drug trial to evaluate effects of Coverscan™, Living with COVID Recovery™, and their interaction, in combination with participants' allocated drug treatment. For all drug therapies, particularly any introduced after start of the trial, comparisons with participants randomised to no drug will include only those participants who could have been randomised to the relevant drug treatment at the time of their randomisation, reducing potential

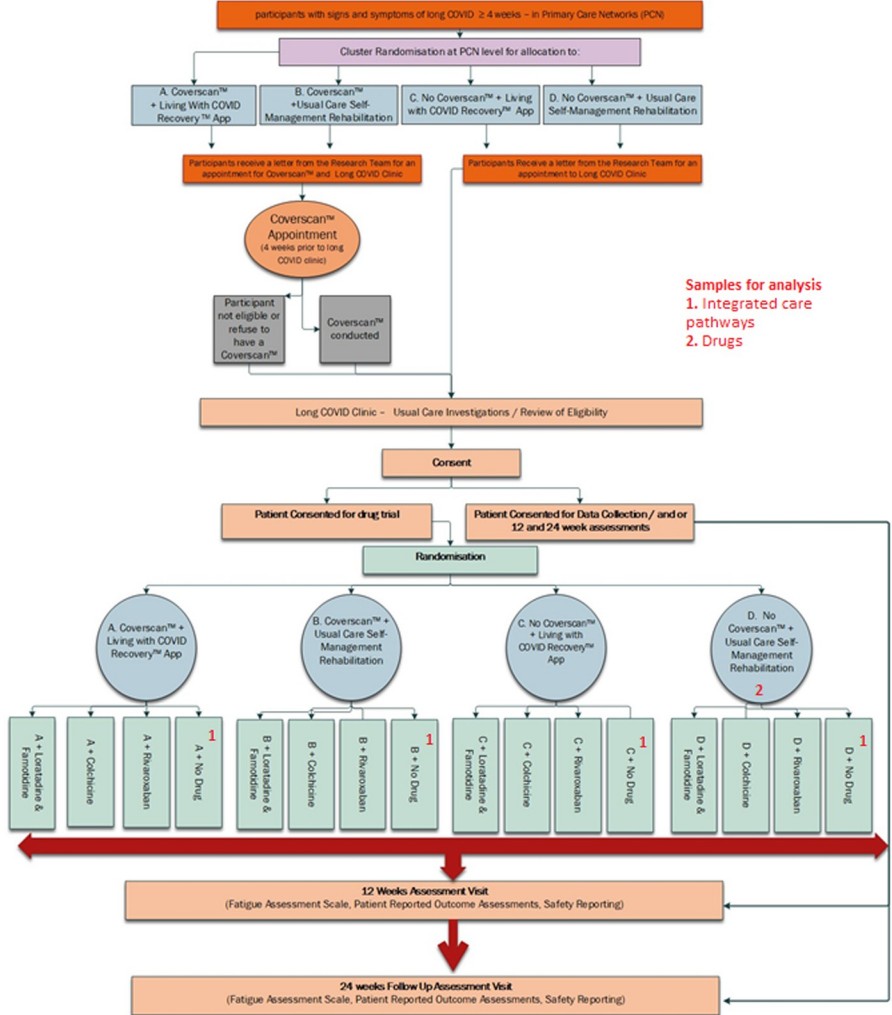

**Fig 3. STIMULATE-ICP study flow.**

bias and ensuring that only concurrent participants allocated to no drug are compared to participants randomised to drug treatments.

## Pathophysiology, trajectory and outcomes, including healthcare utilisation

Using data from blood investigations (e.g. haematology, biochemistry, immunology and -omic analyses), we will study associations with FAS and other outcomes to investigate pathophysiology. We will use linked electronic health record data to monitor healthcare utilisation (e.g. outpatient attendances, emergency department attendances and hospital admissions) and other outcomes (e.g. FAS) in the overall trial population and in subgroups. A separate economic analysis plan will be published and followed.

## Secondary outcomes and missingness

Secondary outcomes on an interval scale will be analysed similarly to the primary outcome, adjusting for baseline outcome measure, clinic and primary care network as a random effect. Where appropriate, alternative methods will be applied to outcomes with skewed distributions.

Patterns of missingness will be summarised. In modelling, complete case analysis will be the main analysis, using a threshold approach as a sensitivity analysis for missing outcome data. Imputation based on thresholds will be implemented such that missing values will be replaced by values from the lower and upper quantiles of the distribution, e.g. 90th quantile, 75th quartile, 25th quartile, 10th quantile, and minimum. Other patient reported secondary outcomes with missing items will be addressed using recommended methods for missing items for that scale.

## Sensitivity and subgroup analyses

A sensitivity analysis of the effects of Coverscan™, Living with COVID Recovery™, and their interaction, will include only those who are randomly allocated to receive usual care (no drug) in the drug trial. Results will be compared to those of the (broader) principal analysis population which also includes those unwilling or unable to be included in the drug trial.

Health outcomes and health utilisation data taken from medical records at 12 months will be described using summary statistics for all participants and for subgroups defined by sub-phenotype (clinical or pathophysiological) at baseline. Analyses of all subgroups will be considered as exploratory. Subgroups may include cardiorespiratory, neuropsychiatric and "mast-cell activation type" (including rashes, joint pain and gastrointestinal disturbances and serositis symptoms). These subgroups are not mutually exclusive as some participants may belong to more than one subgroup depending on their symptoms.

If the Living with COVID Recovery™ application is found to be beneficial, further exploratory analyses will explore factors associated with greater benefit. Some participants allocated to this application will not be able to access it as they may lack a suitable mobile phone or tablet. Other participants will have suitable access but may choose to interact with it on a more or less extensive basis. Amount of access will be available as a patient reported variable and as summary variable from within the application.

## Interim analysis

The first interim analysis point will be after 1200 participants are recruited to the cluster randomised trial and the remaining interim points will be after every 600 participants recruited. The interim analysis time points will apply to both the cluster and drug trials.

Interim analysis of the cluster interventions will take place among participants receiving usual care only (by allocation in the drug trial to no drug or by refusing or being ineligible for the drug trial). A pathway (a combination of the two cluster interventions) could be found to be superior after the first 1200 participants and every 600 participants thereafter in the study. A statistically significant p-value at the 0.001 two-sided level is required for a pathway to be deemed superior at interim analysis. This is to account for multiple testing (Bonferroni type correction). The analysis at 4520 individuals in total will be at the conventional 0.05 two-sided level. There are practical and operational reasons why a single pathway cannot graduate and become the standard pathway within the recruitment period of this trial.

For a drug to graduate early and become part of the backbone treatment of the trial, its two-sided p-value at each of the interim time points will need to be $< 0.001$. The final principal analysis for each drug, including only those participants in the standard of care cluster pathway, will be with at least 200 participants in the active arm and 200 concurrent controls, and will use the conventional 2-sided 0.05 level. This analysis has 85% power if Cohen's D is 0.3 and 97% power if Cohen's D is 0.4.

After consideration and consultation with the IDMC, we decided against futility analysis, partly because it would be both very complicated and very difficult to implement across

different drug arms and different part of the cluster randomised trial, due to our complex study design. Moreover, futility analyses may lead to underpowering of different arms of the trial.

## Power calculation

**Main study.** The sample size was calculated using PASS v21.0.1(2021) for a 2x2 cluster factorial trial. The study was powered to detect an interaction on the FAS scale between Coverscan™ and Living with COVID Recovery™. The sample size does not depend on the size of main effects of both individual interventions. Based on published data the standard deviation on the FAS scale for individuals with LC is estimated at 6 units. For context, the difference in means of FAS between people who say they have or have not recovered from LC was 9 points in a recent observational study [4]. An interaction effect of 3 points [34] on the FAS can be detected with just over 90% power and (two-sided) significance level of 0.05 with 960 participants in 48 primary care network clusters of 20 participants, assuming a conservative intra-cluster correlation coefficient (ICC) of 0.02. If there is dropout of 15% (to be conservative) and assuming missingness is roughly equal across the arms, then the number of participants must be inflated by a factor of 100/85 to 1130 (approximately 56.5 centres of 20 participants).

Experimental estimates of prevalence of symptoms that remain 12-weeks post-COVID range from 3.0% based on tracking specific symptoms, to 11.7% based on self-classification of LC, using data to 1 August 2021 [2]. One LC clinic has ~25 new individuals per week [3], suggesting there might be around 1300 new individuals annually per clinic. Most are expected to give consent to data collection even if not willing to consent to randomisation into a drug study. LC referral rates range from 0–2.5/1000 per primary care network. Assuming this rate will not change over the foreseeable future, this will ensure that our 6–10 LC clinics will provide sufficient patients for this study. Moreover, there is a backlog of people waiting for their first LC clinic appointment (from private communication).

**Nested drug trial.** The total number of individual drugs which will be tested on the platform is currently unknown. The total sample size required in an adaptive platform drug trial, with uncertainty in the timing of the introduction of further drugs, cannot be set in advance, but would need to be updated in consultation with the Independent Data Monitoring Committee (equivalent to the data safety monitoring board) and funder throughout the trial at set interim points in data collection.

The treatment effect is unknown, at this time, for the drugs. The plan is to perform interim analysis after 1200 participants in total are recruited and after every 600 participants thereafter. With 200 participants per active arm and corresponding concurrent controls the study has 85% power to detect a Cohen's D of 0.3 on FAS (difference between means in two treatment arms divided by the standard deviation of the data) at the 0.05 two-sided alpha level. If this number is increased to 300 for the same effect size the power is 95%.

**Planned recruitment rate.** To plan and cost the trial a nominal maximum of 4520 recruited participants was proposed based on up to 4 (drug and usual care) parallel arms of up to 1130 evaluable participants. The nested drug trial aims to recruit up to 4520 individuals from a possible pool of 30 000 individuals from 6–10 primary care network areas of England. The trial will continue to recruit participants at least until the recruitment target of 1130 participants for each of the initial arms of the adaptive trial has been reached. If the adaptive trial requires the full 4520 participants, it is estimated it will take approximately 10 to 12 months for the recruitment to complete and by the 16th month from the start of the trial, the last visit of the last patient would be completed in the third quarter of 2023. The projected recruitment rate, when all clinics are open to recruitment, is 600 participants per month.

## End of trial

The study is deemed to have ended following the last data collection point within the trial. This will be 12 months following enrolment of the last participant in the trial. Thereafter, there will be a 6-month period for data analysis and reporting. The chief investigator and/or trial steering committee have the right at any time to terminate the trial for clinical or administrative reasons. The funder may withdraw funding in the event of futility, study misconduct, or other unanticipated events. In such instances, the Sponsor will notify the Medicines and Healthcare products Regulatory Agency (MHRA) within 15 days. The end of the trial will be reported to the Research Ethics Committee and Regulatory Authority within the required timeframe if the trial is terminated prematurely. Investigators will inform patients of any premature termination of the trial and ensure that the appropriate follow up is arranged for all involved. Following the end of the trial a summary report of the trial will be provided to the Research Ethics Committee and Regulatory Authority within the required timeframe. The Sponsor will notify MHRA at the end of the clinical trial within 90 days of its completion.

The trial may be stopped before completion on the recommendation of the Trial Steering Committee, Independent Data Monitoring or the Sponsor and chief investigator. All safety data will be reviewed and a decision on continuation will be made by the Independent Data Monitoring Committee with input from the Sponsor. A single drug study arm may be stopped by the Independent Data Monitoring Committee on grounds of harm at each stopping point. The Independent Data Monitoring Committee may also recommend a single drug study arm be halted in the event of the death of a single participant, directly attributable to the study drug.

At the end of the trial, all essential documentation and the trial dataset will be prepared for archiving and transfer to the Sponsor by Lancashire Clinical Trials Unit. Participating sites will be asked to archive for a minimum of 25 years from the declaration of end of trial. Essential documents are those which enable both the conduct of the trial and the quality of the data produced to be evaluated and show whether the site complied with the principles of Good Clinical Practice and all applicable regulatory requirements. The Sponsor will notify sites when trial documentation can be archived. All archived documents must continue to be available for inspection by appropriate authorities upon request.

## Oversight

The Trial Management Group is responsible for the management of STIMULATE-ICP and is led by AB (chief investigator) and MH (lead clinical principal investigator). In addition, the trial management group comprises trial investigators and relevant staff from Lancashire Clinical Trials Unit and PPI members, who meet regularly to discuss the progress of the trial. Lancashire Clinical Trials Unit is responsible for day-to-day trial management, is the data custodian, and will conduct central and on-site monitoring of sites and data. STIMULATE-ICP is managed according to Good Clinical Practice guidelines. Lancashire Clinical Trials Unit will act to preserve patient confidentiality and prevent disclosure or reproduction of identifiable patient information by encryption to ensure anonymity. Procedures for handling, processing, storing and destroying data are compliant with the Data Protection Act 1998. The Trial Steering Committee and Independent Data Monitoring Committee have been convened as trial oversight committees. The Trial Steering Committee is independently chaired by Professor Patrick Mallon and includes clinicians, academics, and PPI representatives, along with CI and lead clinical principal investigator. The Trial Steering Committee monitors trial progress. The Independent Data Monitoring Committee is chaired by Professor Peter Langhorne, includes both clinical and statistical expertise, and advises the Trial Steering Committee,

independent of the trial team, sponsor and Trial Steering Committee, making recommendations on continuation, or not, of the trial. Membership of each committee depends on accepting terms of reference and declaration of conflict of interest. Any major protocol modifications (such as changes to eligibility criteria, outcomes, analyses) will be communicated to trial registry (ISRCTN), research team, Research Ethics Committee and Health Research Authority and the public via the study website.

## Ethical approval

The Protocol (version 2.1; 5/5/2022) and other trial-related documentation (and any amendments) received favourable ethical opinion from NRES Committee South Central—Berkshire Research Ethics Committee (reference: 21/SC/0416). All participating sites obtained local approvals prior to patient recruitment.

## Discussion

In individuals with non-hospitalised LC, the STIMULATE-ICP trial represents the first major, pragmatic trial to study: (i) drugs in a nested platform design, (ii) components of an integrated care pathway from pre- to post-clinic care, and (iii) fatigue as a primary endpoint. Given the national and international burden of LC, STIMULATE-ICP is important to investigate feasible, generalisable and scalable improvements in care through: (i) widely-used, re-purposable drugs, (ii) accessible interventions (Coverscan™ and Living with COVID Recovery™), and (iii) study sites varying in geography and clinical service. This paper presents the protocol (Version 2.1, 05/05/2022) and the full protocol is available on the NIHR and study (https://www.stimulate-icp.org/) websites. The SPIRIT checklist [35] for the trial protocol is included in **S9 Appendix**.

Recruitment started in August 2022. At the time of first manuscript submission, data collection for the trial was ongoing and due to be complete in July 2023. The full trial results will be disseminated in Q3 2023 through presentations at national and international conferences, publication in peer reviewed medical journals and regular interaction with media (mainstream and social). As well as the non-trial aspects of STIMULATE-ICP (described in the methods), there are now several nationally funded observational research programmes are underway, including LOCOMOTION, TLC and PC-COS studies [17]. In addition, there are now several ongoing clinical trials in LC:

Phase 1/2

RSLV-132 (https://clinicaltrials.gov/ct2/show/NCT04944121)

Zofin (https://clinicaltrials.gov/ct2/show/NCT05228899)

Axcella-1125 (https://www.clinicaltrials.gov/ct2/show/NCT05152849)

Phase 3

HEAL-COVID (https://heal-covid.net/)

However, to our knowledge, ongoing studies are neither platform studies nor specifically focused on individuals with non-hospitalised LC.

STIMULATE-ICP will provide up-to-date information about investigation and rehabilitation components of the clinical pathway for individuals with LC, as well as developing the evidence for therapeutics in this patient population.

## Supporting information

**S1 File.**
(DOCX)

**S1 Appendix.**
(DOCX)

**S2 Appendix.**
(DOCX)

**S3 Appendix.**
(DOCX)

**S4 Appendix.**
(DOCX)

**S5 Appendix.**
(DOCX)

**S6 Appendix.**
(DOCX)

**S7 Appendix.**
(DOCX)

**S8 Appendix.**
(DOCX)

**S9 Appendix.**
(DOCX)

## Acknowledgments

STIMULATE-ICP team:
    University College London, London, UK:
    Professor Amitava Banerjee (Lead: ami.banerjee@ucl.ac.uk)
    Professor Elizabeth Murray
    Dr Hakim-Moulay Dehbi
    Professor Hugh Montgomery
    Mrs Sarah Clegg
    Dr Henry Goodfellow
    Dr Mel Ramasawmy
    Dr Yi Mu
    Dr Sampath Weerakkody
    Dr Ileana Selejan
    Dr David Sunkersing
    Dr Ashkan Dashtban
    University of Auckland, Auckland, New Zealand
    Professor Paula Lorgelly (also at University College London)
    University of Central Lancashire, Preston, UK:
    Professor Dame Caroline Leigh-Watkins
    Mrs Denise Forshaw
    Dr Gordon Prescott

Royal College of General Practitioners, London, UK:
Dr Gail Allsopp
University of Liverpool, Liverpool, UK:
Professor Mark Gabbay
Professor Gregory Lip
Professor Dan Cuthbertson
Dr Dan Wootton
Professor Nefyn Williams
University of Hull, Hull, UK:
Dr Mike G Crooks
Hull University Teaching Hospitals Trust, Hull, UK:
Dr Angela Green
University of York, York, UK:
Professor Christina Feltz van der Feltz-Cornelis
Dr Jenny Sweetman
Dr Han-I Wang
Ms Natalie Smith
University of Leicester, Leicester, UK:
Professor Kamlesh Khunti
Dr Lauren O'Mahoney
Dr Rachael Evans
University of Exeter, Exeter, UK:
Dr William D Strain
Royal Devon University Healthcare NHS Foundation Trust, Exeter, UK:
Dr Rachel Botell
University of Southampton, Southampton, UK:
Professor Nisreen Alwan,
Dr Donna Clutterbuck
University of Sussex, Brighton, UK:
Dr Marija Pantelic
Living with Covid Recovery:
Mr Chris Robson
Perspectum, Oxford, UK:
Professor Sir Mike Brady
Dr Rajarshi Banerjee
Dr Cat Kelly
Dr Angela Barone
Dr Johannes Alberts
Dr Rob Suriano
STIMULATE-ICP PPI team, University College London, London, UK:
Mr Lyth Hishmeh
Dr Emily Attree
Ms Jasmine Hayer
Ms Rita Mallinson Cookson
Ms Rachel Hext
Mr Andrew Williams
Mrs Rachel Williams
Ms Mag Leahy
Mr Antony Loveless

Mrs Clare Loveless,

Ms Kim Horstmanshof

Collaborators:

Royal College of Speech and Language Therapists: Gemma Clunie

NNEdPro Global Centre for Nutrition and Health: Dominic Crocombe, Shane McAuliffe

Independent Data Monitoring Committee: Peter Langhorne (chair), Chris Sutton

Trial Steering Committee: Independent members: Patrick Mallon (chair), Marion Mafham, Ondine Sherwood (PPI). Other members: Matthew Sydes

UCL/UCLH Joint Research Office (sponsor): Michelle Quaye, Farhat Gilani, Yusuf Jaami, Nikki Cleary, Judy Jones, Robin Holas

## Author Contributions

**Conceptualization:** Denise Forshaw, Emma C. Wall, Gordon Prescott, Caroline Watkins, Melissa Heightman, Amitava Banerjee.

**Data curation:** Gordon Prescott.

**Formal analysis:** Gordon Prescott, Hakim-Moulay Dehbi, Paula Lorgelly.

**Funding acquisition:** Emma C. Wall, Amitava Banerjee.

**Investigation:** Angela Green, Lyth Hismeh, William D. Strain, Michael G. Crooks, Chris Robson, Rajarshi Banerjee, Paula Lorgelly, Melissa Heightman.

**Methodology:** Denise Forshaw, Emma C. Wall, Gordon Prescott, Hakim-Moulay Dehbi, Chris Robson, Melissa Heightman, Amitava Banerjee.

**Project administration:** Denise Forshaw, Emily Attree, Amitava Banerjee.

**Resources:** Amitava Banerjee.

**Supervision:** Denise Forshaw, Emma C. Wall, Hakim-Moulay Dehbi, Emily Attree, Lyth Hismeh, Amitava Banerjee.

**Validation:** Gordon Prescott, Hakim-Moulay Dehbi.

**Writing – original draft:** Amitava Banerjee.

**Writing – review & editing:** Denise Forshaw, Emma C. Wall, Gordon Prescott, Hakim-Moulay Dehbi, Angela Green, Emily Attree, Lyth Hismeh, William D. Strain, Michael G. Crooks, Caroline Watkins, Chris Robson, Rajarshi Banerjee, Paula Lorgelly, Melissa Heightman, Amitava Banerjee.

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
