## [Decision Letter · Decision Letter 0]

11 Sep 2022

PONE-D-22-20207STIMULATE-ICP: A pragmatic, multi-centre, cluster randomised trial of an integrated care pathway with a nested, Phase III, open label, adaptive platform randomised drug trial in individuals with Long COVID: a structured protocol

PLOS ONE

Dear Dr. Banerjee,

Thank you for submitting your manuscript to PLOS ONE. After careful consideration, we feel that it has merit but does not fully meet PLOS ONE’s publication criteria as it currently stands. Therefore, we invite you to submit a revised version of the manuscript that addresses the points raised during the review process.

The manuscript has been evaluated by one reviewer, and their comments are available below.

The reviewer has raised concerns regarding the reporting and conceptualization of the study. 

Could you please revise the manuscript to carefully address the concerns raised?

We look forward to receiving your revised manuscript.

Kind regards,

Johannes Stortz

Staff Editor

PLOS ONE

Journal Requirements:

a) Did participants provide their written or verbal informed consent to participate in this study?

4. Thank you for stating the following in the Funding Section of your manuscript: 

"This trial is part of a National Institute for Health Research (NIHR: COV-LT2-0043)-funded programme (Figure 2)(17), with epidemiologic and mixed methods studies(18), including care inequalities and transferability to other LTCs (IRAS 303958) and this complex intervention trial (IRAS 1004698). Funding for the transport, storage, processing and biobanking of blood and MRI is supported by Perspectum Ltd. The funders had and will not have a role in study design, data collection and analysis, decision to publish, or preparation of the manuscript. The views and opinions expressed therein are those of the authors and do not necessarily reflect those of the NIHR, NHS, the Department of Health and Social Care, or the sponsor."

"This trial is part of a National Institute for Health Research-funded programme  (NIHR: COV-LT2-0043). The CI is AB.

Funding for the transport, storage, processing and biobanking of blood and MRI is supported by Perspectum Ltd (CI-AB). 

The funders had and will not have a role in study design, data collection and analysis, decision to publish, or preparation of the manuscript."

"I have read the journal's policy and the authors of this manuscript have the following competing interests: 

WDS holds research grants from Bayer, Novo Nordisk and Novartis and has received speaker honoraria from AstraZeneca, Bayer, Bristol-Myers Squibb, Merck, Napp, Novartis, Novo Nordisk and Takeda, outside the submitted work. WDS is supported by the NIHR Exeter Clinical Research Facility and the NIHR Collaboration for Leadership in Applied Health Research and Care (CLAHRC) for the South West Peninsula. MGC has received honoraria, fees for advisory boards, and non-financial support from AstraZeneca, BI, Chiesi, GSK, Novartis, and Pfizer and grants from AstraZeneca, BI, Chiesi, and Pfizer. CR is director of Living With, which developed Living With COVID RecoveryTM and other digital health interventions used in healthcare systems including the NHS. RB is CEO of Perspectum, which developed CoverscanTM. AB has received research funding from NIHR, British Medical Association, Astra Zeneca, National Institute of Aging and European Union. AB is a Trustee of Long COVID SoS. All other authors report no competing interests."

6. One of the noted authors is a group or consortium STIMULATE-ICP trial team. In addition to naming the author group, please list the individual authors and affiliations within this group in the acknowledgments section of your manuscript. Please also indicate clearly a lead author for this group along with a contact email address.

Reviewers' comments:

Reviewer's Responses to Questions

**Comments to the Author**

1. Does the manuscript provide a valid rationale for the proposed study, with clearly identified and justified research questions?

Reviewer #1: Yes

2. Is the protocol technically sound and planned in a manner that will lead to a meaningful outcome and allow testing the stated hypotheses?

Reviewer #1: Yes

3. Is the methodology feasible and described in sufficient detail to allow the work to be replicable?

Reviewer #1: Yes

4. Have the authors described where all data underlying the findings will be made available when the study is complete?

Reviewer #1: Yes

5. Is the manuscript presented in an intelligible fashion and written in standard English?

Reviewer #1: Yes

6. Review Comments to the Author

You may also provide optional suggestions and comments to authors that they might find helpful in planning their study.

Reviewer #1: This felt long (at about 6,500 words) and almost a full protocol. My main suggestions are to:

Abbreviate for the journal manuscript to focus on the main design features. The regulatory and logistical aspects could be left out and the current manuscript provided more as an appendix.

In line with this, there were many abbreviations that may be familiar to those in the NIHR context but are not to international readers. I suggest the authors look through the manuscript and spell out most of these abbreviations for readability. An example is what does WP mean in Figure 2?

I was also left with little insight into the actual interventions and the background to those. I would like more details about each intervention - description of what they are exactly, brief background literature review, rationale for choosing these. For example, what is the Coverscan and how is it thought that a multi-organ MRI will help patients?

In the analysis section, it would be helpful if a figure could be provided that illustrates the various different populations for analyses. Either as a stand-alone figure, or indicated on Figure 3.

7. PLOS authors have the option to publish the peer review history of their article (what does this mean?). If published, this will include your full peer review and any attached files.

Reviewer #1: No

---

## [Author Response · Author response to Decision Letter 0]

5 Oct 2022

We have responded to reviewers in the attached letter.

---

## [Decision Letter · Decision Letter 1]

1 Dec 2022

PONE-D-22-20207R1STIMULATE-ICP: A pragmatic, multi-centre, cluster randomised trial of an integrated care pathway with a nested, Phase III, open label, adaptive platform randomised drug trial in individuals with Long COVID: a structured protocolPLOS ONE

Dear Dr. Amitava,

Thank you for submitting your manuscript to PLOS ONE. After careful consideration, we feel that it has merit but does not fully meet PLOS ONE’s publication criteria as it currently stands. Therefore, we invite you to submit a revised version of the manuscript that addresses the points raised during the review process.

We look forward to receiving your revised manuscript.

Kind regards,

Hideo Kato

Academic Editor

PLOS ONE

Journal Requirements:

Additional Editor Comments:

The authors should take seriously in account the comments of the reviewers.

Reviewers' comments:

Reviewer's Responses to Questions

**Comments to the Author**

1. Does the manuscript provide a valid rationale for the proposed study, with clearly identified and justified research questions?

Reviewer #1: Yes

Reviewer #2: Yes

2. Is the protocol technically sound and planned in a manner that will lead to a meaningful outcome and allow testing the stated hypotheses?

Reviewer #1: Yes

Reviewer #2: Yes

3. Is the methodology feasible and described in sufficient detail to allow the work to be replicable?

Reviewer #1: Yes

Reviewer #2: Yes

4. Have the authors described where all data underlying the findings will be made available when the study is complete?

Reviewer #1: Yes

Reviewer #2: No

5. Is the manuscript presented in an intelligible fashion and written in standard English?

Reviewer #1: Yes

Reviewer #2: Yes

6. Review Comments to the Author

You may also provide optional suggestions and comments to authors that they might find helpful in planning their study.

Reviewer #1: Thank you, the manuscript is much improved.

I still think there needs to be additional rationale provided for use of Coverscan. The authors have still not addressed the previous question sufficiently. It is not clear to me how a multi-organ MRI will result in improved outcomes. Will there be an algorithmic management approach based on the findings? For example, what is proposed if there is pancreatic or renal involvement in the absence of other markers of organ dysfunction? I appreciate the work showing that 70% of a previous cohort had evidence of at least single organ involvement, what is not clear is whether this has any clinical implications and how it might result in changes in management that would ultimately benefit the patient.

Reviewer #2: No mentioning of data safety monitoring board.

Futility analysis was not considered in the interim analysis.

P value <0.001 is not clearly justified in the interim analysis.

7. PLOS authors have the option to publish the peer review history of their article (what does this mean?). If published, this will include your full peer review and any attached files.

Reviewer #1: No

Reviewer #2: No

---

## [Author Response · Author response to Decision Letter 1]

8 Dec 2022

RESPONSE TO REVIEWERS

PONE-D-22-20207R1

STIMULATE-ICP: A pragmatic, multi-centre, cluster randomised trial of an integrated care pathway with a nested, Phase III, open label, adaptive platform randomised drug trial in individuals with Long COVID: a structured protocol

PLOS ONE

Dear Dr. Amitava,

Thank you for submitting your manuscript to PLOS ONE. After careful consideration, we feel that it has merit but does not fully meet PLOS ONE’s publication criteria as it currently stands. Therefore, we invite you to submit a revised version of the manuscript that addresses the points raised during the review process.

Thanks- we have included these documents in the revised submission.

Our protocol is already in the preprint repository (Medrxiv): https://www.medrxiv.org/content/10.1101/2022.07.21.22277893v1. Our trial is also registered at ISRCTN https://www.isrctn.com/ISRCTN10665760

We look forward to receiving your revised manuscript.

Kind regards,

Hideo Kato

Academic Editor

PLOS ONE

Journal Requirements:

Additional Editor Comments:

The authors should take seriously in account the comments of the reviewers.

Reviewers' comments:

Reviewer's Responses to Questions

Comments to the Author

1. Does the manuscript provide a valid rationale for the proposed study, with clearly identified and justified research questions?

Reviewer #1: Yes

Reviewer #2: Yes

Thanks

2. Is the protocol technically sound and planned in a manner that will lead to a meaningful outcome and allow testing the stated hypotheses?

Reviewer #1: Yes

Reviewer #2: Yes

Thanks

3. Is the methodology feasible and described in sufficient detail to allow the work to be replicable?

Reviewer #1: Yes

Reviewer #2: Yes

Thanks

4. Have the authors described where all data underlying the findings will be made available when the study is complete?

Reviewer #1: Yes

Reviewer #2: No

Thanks- it is unclear why Reviewer 2 has written “No” as we do not provide results/findings in this protocol (since the trial is currently recruiting). We have provided data supporting the choice of drugs as supporting (supplementary) information. See response below.

5. Is the manuscript presented in an intelligible fashion and written in standard English?

Reviewer #1: Yes

Reviewer #2: Yes

Thanks

6. Review Comments to the Author

You may also provide optional suggestions and comments to authors that they might find helpful in planning their study.

Reviewer #1: Thank you, the manuscript is much improved.

I still think there needs to be additional rationale provided for use of Coverscan. The authors have still not addressed the previous question sufficiently. It is not clear to me how a multi-organ MRI will result in improved outcomes. Will there be an algorithmic management approach based on the findings? For example, what is proposed if there is pancreatic or renal involvement in the absence of other markers of organ dysfunction? I appreciate the work showing that 70% of a previous cohort had evidence of at least single organ involvement, what is not clear is whether this has any clinical implications and how it might result in changes in management that would ultimately benefit the patient.

Thanks for this remark and apologies if the previous response was not clear. Part of the trial is to understand how the Coverscan may inform or influence care (or not), and it came on the backdrop of high numbers of patients seeking investigation for possible Long Covid with variable waiting times, variable resources for scanning and variable practices with frequent repeating of investigations and referrals to multiple specialists (see reference 3. Heightman M et al. 2021). A multi-organ MRI may improve outcomes in two ways, both by reducing time to treatment. First, a negative finding by ruling out organ involvement and other disease early could reassure the clinician and the patient, rather than organising multiple investigations with multiple specialists via multiple referrals. Second, by flagging organ or multi-organ dysfunction, the MRI may trigger appropriate specialist referral earlier and lead to earlier appropriate management. Of course, the trial may show no effect of Coverscan.

The reviewer is right that we do not know yet the clinical implications of the single or multiple organ involvement post-COVID and the rate of resolution- these are being investigated in the trial. Therefore, we did not have or develop an algorithmic management approach (i.e. clinical decision support) or “manual” because the organ dysfunction in Long Covid is still being defined and a manual would involve approaches to any possible finding in lungs, heart, liver, pancreas and kidneys, which is not feasible. In answer to the reviewer’s question, “…what is proposed if there is pancreatic or renal involvement in the absence of other markers of organ dysfunction?”, there would be specialist referral to assess the potential relevance or importance of such a finding, which may be incidental and not clinically relevant.

The Coverscan has a standardised report. For the first 100 scans across the 6 sites, there is report by an independent radiologist, and a virtual, weekly multidisciplinary meeting to discuss any cases and potential implications of findings, management and/or referral to specialists. To summarise, part of the trial is to find out whether the Coverscan changes and/or improves management of people with Long Covid. 

We have added text to the revised manuscript to cover some of these complex issues.

Reviewer #2: No mentioning of data safety monitoring board.

We do mention the Independent Data Monitoring Committee which is the data safety monitoring board (we have now explained this in the new revised text). 

Futility analysis was not considered in the interim analysis.

Thank you for this important remark. We actually had considered futility analysis in the interim analysis and our study but both our team and the Independent Data Monitoring Committee felt that futility analysis should not be included because:

1. Our focus is on drugs and integrated pathway components which work

2. Due to the complex design, a futility analysis plan across different drug arms and different part of the cluster randomised trial was likely to be complicated and very difficult to implement.

3. In a trial with several moving parts and a moving target in terms of the disease itself, futility analyses may lead to underpowering of arms of the study. 

We and our Independent Data Monitoring Committee (who we consulted again) feel that the plans for interim analysis are robust. 

We have added a sentence regarding futility analysis to reflect these issues in the revised manuscript. 

P value <0.001 is not clearly justified in the interim analysis.

We do have a sentence in the manuscript already to justify the p value<0.001: 

“A statistically significant p-value at the 0.001 two-sided level is required for a pathway to be deemed superior at interim analysis. This is to account for multiple testing (Bonferroni type correction). The analysis at 4520 individuals in total will be at the conventional 0.05 two-sided level. There are practical and operational reasons why a single pathway cannot graduate and become the standard pathway within the recruitment period of this trial.”________________________________________

7. PLOS authors have the option to publish the peer review history of their article (what does this mean?). If published, this will include your full peer review and any attached files.

Do you want your identity to be public for this peer review? For information about this choice, including consent withdrawal, please see our Privacy Policy.

Reviewer #1: No

Reviewer #2: No

 Thanks.

---

## [Decision Letter · Decision Letter 2]

1 Feb 2023

STIMULATE-ICP: A pragmatic, multi-centre, cluster randomised trial of an integrated care pathway with a nested, Phase III, open label, adaptive platform randomised drug trial in individuals with Long COVID: a structured protocol

PONE-D-22-20207R2

Dear Dr. Amitava Banerjee,

We’re pleased to inform you that your manuscript has been judged scientifically suitable for publication and will be formally accepted for publication once it meets all outstanding technical requirements.

Kind regards,

Hideo Kato

Academic Editor

PLOS ONE

Additional Editor Comments (optional):

Reviewers' comments:

Reviewer's Responses to Questions

**Comments to the Author**

1. Does the manuscript provide a valid rationale for the proposed study, with clearly identified and justified research questions?

Reviewer #1: Yes

Reviewer #2: Yes

2. Is the protocol technically sound and planned in a manner that will lead to a meaningful outcome and allow testing the stated hypotheses?

Reviewer #1: Yes

Reviewer #2: Yes

3. Is the methodology feasible and described in sufficient detail to allow the work to be replicable?

Reviewer #1: Yes

Reviewer #2: Yes

4. Have the authors described where all data underlying the findings will be made available when the study is complete?

Reviewer #1: Yes

Reviewer #2: Yes

5. Is the manuscript presented in an intelligible fashion and written in standard English?

Reviewer #1: Yes

Reviewer #2: Yes

6. Review Comments to the Author

You may also provide optional suggestions and comments to authors that they might find helpful in planning their study.

Reviewer #1: Thanks for the responses. I am satisfied with the changes.

AAAAaaaaaaaaaaaaaaaaaaaaaaaaaaaaaaaaaaaa (satisfying minimum character count)

Reviewer #2: All my concerns were addressed.

7. PLOS authors have the option to publish the peer review history of their article (what does this mean?). If published, this will include your full peer review and any attached files.

Reviewer #1: No

Reviewer #2: No

---

## [Editor Report · Acceptance letter]

5 Feb 2023

PONE-D-22-20207R2 

STIMULATE-ICP: A pragmatic, multi-centre, cluster randomised trial of an integrated care pathway with a nested, Phase III, open label, adaptive platform randomised drug trial in individuals with Long COVID: a structured protocol 

Dear Dr. Banerjee:

I'm pleased to inform you that your manuscript has been deemed suitable for publication in PLOS ONE. Congratulations! Your manuscript is now with our production department. 

Kind regards, 

on behalf of

Dr. Hideo Kato 

Academic Editor

PLOS ONE